# Overexpression of Growth Differentiation Factor 15 in Glioblastoma Stem Cells Promotes Their Radioresistance

**DOI:** 10.3390/cancers16010027

**Published:** 2023-12-20

**Authors:** Alexandre Bentaberry-Rosa, Yvan Nicaise, Caroline Delmas, Valérie Gouazé-Andersson, Elizabeth Cohen-Jonathan-Moyal, Catherine Seva

**Affiliations:** 1Centre de Recherche en Cancérologie de Toulouse (CRCT), INSERM U1037, Université Toulouse III Paul Sabatier, ERL5294 CNRS, 31062 Toulouse, France; bentaberryrosa.alexandre@iuct-oncopole.fr (A.B.-R.); yvan.nicaise@inserm.fr (Y.N.); delmas.caroline@iuct-oncopole.fr (C.D.); valerie.gouaze-andersson@inserm.fr (V.G.-A.); moyal.elizabeth@iuct-oncopole.fr (E.C.-J.-M.); 2IUCT-Oncopole, 31100 Toulouse, France

**Keywords:** glioblastomas, radioresistance, signaling, GDF15, cancer stem cells

## Abstract

**Simple Summary:**

Glioblastoma stem cells (GSCs) are characterized by high radioresistance and are responsible for the frequent recurrence of glioblastoma (GBM); therefore, it is imperative to better understand the molecular mechanisms and identify the novel factors involved in these processes. Here, we report that GDF15, a cytokine belonging to the TGF-β superfamily, is induced by irradiation (IR) and the transcription factor, WWTR1/TAZ. It also promotes GSC radioresistance. These data suggest that blocking GDF15 may be a novel approach in radiosensitizing GSCs.

**Abstract:**

GSCs play an important role in GBM recurrence. Understanding the resistance mechanisms in these cells is therefore crucial for radiation therapy optimization. In this study, using patient-derived GSCs, we demonstrate that GDF15, a cytokine belonging to the TGF-β superfamily, is regulated by irradiation (IR) and the transcription factor WWTR1/TAZ. Blocking WWTR1/TAZ using specific siRNAs significantly reduces GDF15 basal expression and reverses the upregulation of this cytokine induced by IR. Furthermore, we demonstrate that GDF15 plays an important role in GSC radioresistance. Targeting GDF15 expression by siRNA in GSCs expressing high levels of GDF15 sensitizes the cells to IR. In addition, we also found that GDF15 expression is critical for GSC spheroid formation, as GDF15 knockdown significantly reduces the number of GSC neurospheres. This study suggests that GDF15 targeting in combination with radiotherapy may be a feasible approach in patients with GBM.

## 1. Introduction

Glioblastoma (GBM) is an aggressive tumor of the central nervous system, characterized by rapid growth and high infiltration, making surgical resection difficult and incomplete. Despite radiotherapy and adjuvant chemotherapy, the median overall survival is around 15 months [1]. The GSCs that are particularly resistant to treatment and highly invasive are involved in the recurrence of GBM [2,3]; therefore, understanding the resistance mechanisms in GSCs is important to identify new therapeutic approaches.

GDF15 (growth differentiation factor 15) is a cytokine that belongs to the TGF-β superfamily. Its role in cancer appears contradictory, with an anti-tumoral effect in the early stages of tumor development, but a pro-tumoral effect in the later stages [4]. GDF15 is overexpressed in many cancers (colorectal, gastric, pancreatic, hepatic, ovarian) and is often associated with tumor aggressiveness [5,6,7,8,9,10]. Its expression can be induced by cellular stress conditions or the tumor microenvironment [11,12,13]. Notably, GDF15 expression can also be induced following IR, hypoxia, or by tumor-associated macrophages.

In glioma, GDF15 expression increases with the tumor grade. High levels of GDF15 in the tumor or in the cerebrospinal fluid are associated with a poor prognosis for GBM patients [14,15,16]. GDF15 also plays a role in the invasive capacity and proliferation of GBM cell lines [17,18,19]. In addition, GDF15 is correlated with the inflammatory response of glioma and may contribute to an immunosuppressive micro-environment, as well as an immune escape of GBM. Interestingly, the regulation of PDL1 expression by GDF15, through the smad2/3 pathway, has been reported in several GBM cell lines [14,19,20].

The expression and role of GDF15 have been little studied in GSCs. To our knowledge, only one publication reports on a cancer stem cell line from a single patient, reporting an increase in neurosphere formation in response to the addition of exogenous GDF15 [21]. In this study, we analyzed the expression of GDF15 in GSCs derived from 13 human GBM biopsy specimens, as well as the function of GDF15 in GSCs.

We report that GDF15 is increased after IR and regulated by the transcription factor, WWTR1/TAZ, in GSCs. We also demonstrate that targeting GDF15 decreases neurosphere formation and increases the radiosensitivity of GSCs, suggesting that GDF15 is an attractive therapeutic target to radiosensitize GBM.

## 2. Materials and Methods

### 2.1. GSCs Derived from GBM Biopsy Specimens

GBM biopsies were obtained from the Department of Neurosurgery, Toulouse University Hospital. This clinical study (PI Pr. E. Cohen-Jonathan-Moyal) was approved by the Human Research Ethics Committee (Ethics Code 12TETE01, ID-RCB No. 2012-A00585-38, Approval Date: 7 May 2012). Written informed consent of all patients was obtained. According to the World Health Organization, all the tumors were classified as GBM. The primary neurospheres of GSCs were obtained from GBM samples, as described by Avril [14], and cultured in DMEM-F12 (GIBCO, New York, NY, USA) containing B27 and N2 (Life-Technologies, Carlsbad, CA, USA), FGF-2, and EGF (PeproTech, Rocky Hill, NJ, USA). The GSCs in this study have been previously described and validated for their self-renewal capacity, the expression of stem cell markers, and their ability to differentiate into neural lineages and develop tumors in vivo [22,23,24,25]; Appendix A shows the stem cell marker expression. Neurospheres were passaged less than 12 times to maintain cell properties.

### 2.2. RNA Sequencing

RNA was extracted from GSCs using the RNeasy kit (Qiagen, Venlo, The Netherlands) and quantified using the RNA Qubit Broad Range Kit (Thermo Fisher Scientific, Waltham, MA, USA). Purity was verified using the NanoDrop ND-100 (Thermo Fisher Scientific) spectrophotometer. The RNA quality was checked using a fragment analyzer (Agilent Technology, Santa Clara, CA, USA) to obtain “RIN” (RNA integrity number) and verify the absence of genomic DNA. For library preparation, 400 ng of total RNA was used. NGS-based RNA sequencing was performed on a Nextseq 550 instrument using Illumina Stranded Total RNA Prep, and ligation with Ribo-Zero Plus Kit (Illumina, San Diego, CA, USA) following the manufacturer protocol (Illumina, San Diego, CA, USA).

### 2.3. Bioinformatic Analysis

The mRNA expression of GDF15, WWTR1/TAZ, YAP1, CYR61, and CTGF was compared between normal tissues, GBM, and GSCs. GBM samples and 10 matched adjacent normal tissue samples were downloaded from the TCGA database. The data for GSCs were obtained from the RNAseq that was performed on GSCs derived from 13 human GBM biopsy specimens. All data were normalized with GAPDH expression to compare expression levels. Pairwise t-tests were performed between the groups.

To identify the genes that were significantly and positively correlated with a high expression of GDF15 in GBM patients, we used the Agilent-4502A platform on the TCGA-GBM database of the Gliovis website “http://gliovis.bioinfo.cnio.es (accessed on 20 September 2023)”. The Spearman method was used with a *p*-value < 0.01.

The correlation between GDF15 expression and WWTR1/TAZ, YAP1, CYR61, or CTGF in GBM was obtained using a co-expression analysis in Gliovis “http://gliovis.bioinfo.cnio.es (accessed on 20 September 2023)” using the TCGA database. The correlation between GDF15 expression and WWTR1/TAZ in GSCs was obtained from the RNAseq data, performed on GSCs derived from 13 human GBM biopsy specimens and visualized using a correlation curve performed with Srplot “http://www.bioinformatics.com.cn/srplot (accessed on 20 September 2023)”, an online platform for data analysis and visualization. The links between biomarkers were assessed using Spearman correlation coefficients and their associated *p*-values.

The differentially expressed genes between the GSCs expressing high or low levels of GDF15 were determined using the RNAseq dataset (fold change cut off ≥ 1.7 and *p*-value < 0.05) and visualized using a volcano plot performed with Srplot “http://www.bioinformatics.com.cn/srplot (accessed on 20 September 2023)”, an online platform for data analysis and visualization.

Gene ontology analyses were performed on the genes that were significantly up-regulated in the GSCs expressing high levels of GDF15 using the Metascape platform “https://metascape.org (accessed on 20 September 2023)”, a gene annotation and analysis resource. The main pathways were visualized with an enrichment bubble plot performed with Srplot “http://www.bioinformatics.com.cn/srplot (accessed on 20 September 2023)”, an online platform for data analysis and visualization.

### 2.4. Human XL Proteome Profiler Array (Cytokines Array)

The R&D Human XL Cytokine Array kit (R&D systems, Minneapolis, MN, USA) was used for cytokine detection in the conditioned media of GSCs according to the supplier’s protocol. Briefly, 5 × 10^5^ GSCs were seeded in a T25 flask containing 5 mL of medium. When indicated, cells were transfected with 166 pmol of RNAi (according to the siRNA supplier) as described below. After 48 h, nitrocellulose membranes containing antibodies corresponding to the 105 cytokines were incubated with 500 µL of conditioned media. After incubation with a cocktail of biotinylated antibodies, detection was performed using chemiluminescence with streptavidin-HRP using ChemiDoc XRS+ system and Image Lab software 6.1. For data analysis, the average signal of the pair of duplicate spots was determined and the background signal corresponding to the negative controls was subtracted. The results were normalized to the average signal of the reference spots and the number of cells.

### 2.5. SiRNA Transfection, RNA Extraction, Reverse Transcription and Real-Time PCR

WWTR1/TAZ and GDF15 siRNAs or the scramble control were obtained from Qiagen. Lipofectamine RNAi Max (Invitrogen, Waltham, MA, USA) was used for transfection according to the manufacturer’s protocol. Briefly, 5 × 10^5^ GSCs were seeded in a T25 flask containing 5 mL of medium and transfected with 166 pmol of RNAi according to the supplier. 48 h after transfection, total RNA was extracted with the RNeasy RNA isolation Kit (Qiagen) and reverse transcription was carried out using the Prime Script RT Reagent kit (TAKARA, Kusatsu, Japan). ABI-Stepone+ and StepOne software V2.2.2 (Applied Biosystems, Waltham, MA, USA) was used to perform the real-time PCR, with GAPDH for the normalization. The following RNAi were used: si-WWTR1/TAZ (# 4392420 from Ambion, Foster City, CA, USA), si-GDF15 (1), and si-GDF15 (2) (respectively, SI00069461 and SI00069468 from Qiagen).

### 2.6. Western-Blot Analysis

The following antibodies: Actin (Millipore, Burlington, MA, USA) and WWTR1/TAZ (Cell Signaling Technology, Danvers, MA, USA), were used in the western-blotting analysis performed as previously described [26].

### 2.7. GSCs Irradiations

GSCs were exposed to different doses of IR (2 to 8 Gy), as indicated, using an irradiator SmART+ irradiator (Precision X-ray Inc., Madison, WI, USA).

### 2.8. Neurosphere-Forming Analysis

48 h after transfection with specific GDF15, WWTR1 siRNAs or scramble control cells were seeded in 96-well plates (500 cells/wells, 12 wells per condition) and irradiated or not with increasing doses of X-rays (0 to 6 Gy) using the SmART+ irradiator (Precision X-ray Inc., Madison, WI, USA). The neurospheres/wells were counted 8–10 days post IR under a microscope. Calculation of the surviving fraction (SF) was performed and took into account the plating efficiency (PE = spheres number/seeded cells number × 100) of the non-irradiated condition (SF = PE × Gy/PE 0 gy × 100).

## 3. Results

### 3.1. GDF15 Expression in GBM and GSCs

Using RNAseq data normalized with GAPDH, we compared the expression levels of GDF15 mRNA in normal tissue and GBM samples from the TCGA database with those of GSCs derived from 13 human GBM biopsy specimens. Figure 1A shows a significantly higher expression of GDF15 in GBM and GSCs compared with normal tissue, while no significant difference was observed between GBM and GSCs. We then analyzed the secretion of GDF15 in the conditioned media of GSCs derived from two different patients using the R&D Human XL Cytokine Array kit, which targets human cytokines and growth factors. Figure 1B,C shows that among the 105 cytokines and factors analyzed, GDF15 is well secreted by GSCs into the medium.

### 3.2. GDF15 Expression Is Regulated by the Transcription Factor WWTR1/TAZ in GSCs

We then used the TCGA-GBM database on the Gliovis website to identify the genes in GBM that were positively and significantly (*p*-value < 0.01) correlated with a high expression of GDF15 compared with low expression. The global gene list is provided in Appendix A. Interestingly, we observed that the expression of several genes of the Hippo pathway, including the transcription factor WWTR1 (also known as TAZ), its paralog, YAP1, and their target genes, CYR61 and CTGF, are significantly correlated to GDF15 (Figure 2A). RNAseq data analyses normalized with GAPDH show (Figure 2B) a higher expression of WWTR1/TAZ in GBM and GSCs compared with normal tissue. In contrast, YAP1, Cyr61, and CTGF are overexpressed only in GBM compared with normal tissue.

In the GSCs used in this study, we also observed a positive and significant correlation between WWTR1/TAZ and GDF15 expression (Figure 2C), suggesting a link between these two genes. In contrast, we did not observe a significantly higher expression of YAP1, CYR61, or CTGF in the GSCs expressing high levels of GDF15.

Based on the correlation observed between WWTR1/TAZ and GDF15, and because WWTR1/TAZ is a crucial transcription factor for the aggressiveness and resistance of GSCs [27,28], we made the hypothesis that WWTR1/TAZ might regulate the expression of GDF15 in GSCs. The role of WWTR1/TAZ in the regulation of GDF15 expression has never been reported. To investigate whether WWTR1/TAZ could be involved in this process, we analyzed by RT-QPCR the expression of GDF15 in GSCs derived from two patients, in which WWTR1/TAZ was inhibited using a specific siRNA validated for its efficiency to inhibit WWTR1/TAZ expression compared with a scramble control (Figure 3A,B and Appendix A). As shown in Figure 3C, GSCs deficient for WWTR1/TAZ show a significant inhibition of GDF15 mRNA expression relative to control cells, suggesting a role of this transcription factor in GDF15 regulation. These results are confirmed at the protein level on secreted GDF15 by performing cytokine arrays on the conditioned media of GSCs transfected with a WWTR1/TAZ siRNA or a scramble control. Figure 3D illustrates the decrease in GDF15 in the conditioned media of GSCs transfected with the WWTR1/TAZ siRNA. The relative quantifications of secreted GDF15 (Figure 3E), normalized to the average signal of the reference spots (Appendix A) and the number of cells as described in “Methods”, confirm a significantly reduced expression of GDF15 in GSCs when WWTR1/TAZ is inhibited.

### 3.3. Down-Regulation of GDF15 Gene Expression Decreases Sphere-Forming Ability and Radiosensitizes GSCs

To determine the mechanism by which GDF15 might function in GSCs, we analyzed the genes that were differentially expressed between GSCs with a strong expression of GDF15 compared with GSCs expressing GDF15 weakly. The volcano plot (Figure 4A) displays the differential changes (fold change cut off ≥ 1.7 and *p*-value < 0.05), with the grey dots representing the genes that are not differentially expressed, the blue dots representing those that are significantly down-regulated, and the red dot representing those that are significantly up-regulated. The global list of the genes significantly up-regulated is provided in Appendix A. Next, gene ontology analysis was performed on the genes significantly up-regulated in the high expressing GDF15 GSCs. These results suggest that GDF15 might contribute to cell proliferation in GSCs but also cellular response to stress and DNA damage and repair (Figure 4B); therefore, we hypothesize that GDF15 might be involved in the radioresistance of GSCs.

First, we examined the role of GDF15 in the ability of GSCs to form neurospheres. Sphere formation in non-irradiated samples is significantly reduced when GDF15 expression is blocked in GSC02 with two different GDF15 siRNAs, as demonstrated in Figure 5A–C. These results were confirmed in the GSC07 cells (Appendix A).

It has been previously reported that IR can activate pro-survival signaling pathways and increase the expression of radioresistance-related factors; thus, we analyzed the possibility that IR could stimulate the expression of GDF15 in GSCs. Figure 6A illustrates how IR significantly increases GDF15 expression in GSCs. Figure 6B shows that blocking WWTR1/TAZ by a specific siRNA can reverse IR-induced GDF15 expression, indicating that GDF15 expression induced by IR is dependent on WWTR1/TAZ. On the contrary, WWTR1/TAZ expression is not increased in response to IR.

To analyze the role of GDF15 in radioresistance, we treated the GSC02 transfected with two different GDF15 siRNAs or a scramble control with increasing doses of IR (0 to 6 Gy). The survival fraction after IR is significantly reduced in GSCs expressing specific GDF15 siRNAs compared with the control, indicating that GSCs can be radiosensitized by downregulating GDF15 expression (Figure 6C). These results were confirmed in the GSC07 cells (Appendix A). In addition, the inhibition of cell survival observed at 6 Gy in the cells expressing a specific GDF15 siRNA was rescued with the addition of recombinant human GDF15 (20 ng/mL) before IR (Figure 6D). As expected, the ability of GSCs to form neurospheres as well as radioresistance was decreased when WWTR1/TAZ expression was inhibited (Appendix A).

## 4. Discussion

Tumor stem cells that are particularly invasive and resistant to radiotherapy are mainly responsible for the recurrence of GBM. Currently, no therapy specifically targets GSCs; therefore, it is important to fully understand the molecular mechanisms associated with the resistance of these cells in order to be able to propose new therapeutic approaches. In this study, we focused on the expression of a member of the TGF-β superfamily, GDF15, and its potential role in GSCs. This study is the first one demonstrating that GDF15 is expressed in GSCs and contributes to their radioresistance.

First, we showed that GDF15 was well expressed and secreted by GSCs using GSCs derived from 13 human GBM biopsy specimens. We also demonstrated an increase in GDF15 expression after IR. In addition, we determined one of the molecular mechanisms that regulates GDF15 expression in GSCs. Indeed, we identified WWTR1/TAZ, an oncogenic driver involved in GBM tumorigenicity, invasion, and radioresistance [27,28,29,30], as a regulator of GDF15 expression in GSCs. Blocking WWTR1/TAZ using an RNAi approach decreased both the basal and IR-induced expression of GDF15. This is the first study that shows that WWTR1/TAZ is an upstream regulator of GDF15; however, WWTR1/TAZ is a transcription factor that has been previously shown to regulate the expression of numerous factors and pathways [31]. Therefore, we cannot exclude that the regulation of GDF15 by WWTR1 may be indirect and involve WWRT1-dependent pathways.

In the cancer cell literature, only two publications have reported contradictory effects of the WWTR1/TAZ paralog, YAP, which represses GDF15 expression in breast cancer and induces expression in hepatocellular carcinoma [32,33]. Although WWTR1/TAZ and YAP are paralogs, we did not observe a correlation between GDF15 and YAP expression in GSCs. Blocking YAP expression by specific siRNAs, validated for their efficiency, did not affect GDF15 expression, suggesting a specificity of WWTR1/TAZ in the regulation of GDF15 in GSCs.

In accordance with our results, which showed a regulation of GDF15 expression by IR in the GSCs, it was previously published that IR could also induce the expression and secretion of this cytokine in oral carcinoma cells [34,35]. Similarly, it was shown that IR regulates GDF15 expression by cells in the tumor microenvironment, including macrophages, fibroblasts, or endothelial cells [11,36,37]. In particular, GDF15 secretion by irradiated endothelial cells has been shown to contribute to angiogenesis in GBM models [11].

In addition, we showed that targeting GDF15 in GSCs, which express high levels of this factor, sensitized cells to radiation. To our knowledge, the role of GDF15 in the radioresistance of cancer stem cells, particularly GSCs, has never been reported; however, some publications have shown that high expression of GDF15 contributes to the radioresistance of established cancer cell lines. Knockdown of GDF15 in breast, oral, and head and neck cancer cell lines sensitized resistant cells to IR [34,35,38,39,40]. How GDF15 induces the radioresistance of cancer cells is not well understood. Through bioinformatics analyses, we showed in this study that several signaling pathways associated with DNA damage response and repair were enriched in GSCs expressing high levels of GDF15. In particular, BRCA1, XRCC2, or EXO1 that are linked to these pathways were up-regulated (Appendix A); however, inhibiting GDF15 by specific siRNAs did not affect the expression of these genes, suggesting an indirect mechanism linking GDF15 to these signaling pathways that remains to be elucidated. In head and neck, and breast cancer, it has been reported that the radioresistance induced by GDF15 is linked to the expression of stemness markers (CD44, Sox2, and Nanog) and the ability of cancer cells to form spheroids [35,40]. These publications are in accordance with our results, which show that GDF15 is involved in neurosphere formation by GSCs; however, we did not observe a decrease in stem cell marker expression (CD44, Sox2, Nanog, Nestin) when GDF15 was inhibited by specific siRNAs. These differences suggest that GDF15 may act through distinct signaling pathways depending on the specific type of cancer.

The literature has also reported the role of WWTR1/TAZ in radioresistance. In esophageal, lung, and pancreatic cancer, as well as in GBM, WWTR1/TAZ has been shown to confer resistance to radiation [30,41,42,43]. In accordance with these publications, we also showed in this study the radiosensitization of GSCs by inhibiting WWTR1/TAZ expression. This effect can be partly attributed to GDF15 inhibition when WWTR1/TAZ is blocked but might also involve other factors and pathways regulated by WWTR1/TAZ [31].

Very few bibliographic data have been published on the expression and role of GDF15 in cancer stem cells. In head and neck, breast, and gastric tumors, a high expression of GDF15 by cancer stem cells contributes to maintaining their properties [40,44,45]. In liver cancer, GDF15 produced by the cancer stem cells promotes their proliferation and metastatic power [46]. In addition, GDF15 secreted by bone marrow stromal cells has been reported to contribute to tumor initiation and self-renewal of multiple myeloma cells [47]. In addition to its effect on GSC radioresistance, we have shown that GDF15 expression is crucial for sphere formation by GSCs, since the knockdown of GDF15 significantly decreases the number of neurospheres formed by GSCs derived from GBM biopsy specimens. In accordance with these results obtained in GSCs, Zhu et al. previously showed that the addition of recombinant GDF15 to differentiated U87 cells promoted neurosphere formation [21].

## 5. Conclusions

All of our results demonstrate a novel mechanism of GSC radioresistance in which GDF15 plays a critical role. This also opens a new potential therapeutic strategy involving GDF15 targeting in association with radiation therapy to overcome GBM radioresistance.

## Figures and Tables

**Figure 1 cancers-16-00027-f001:**
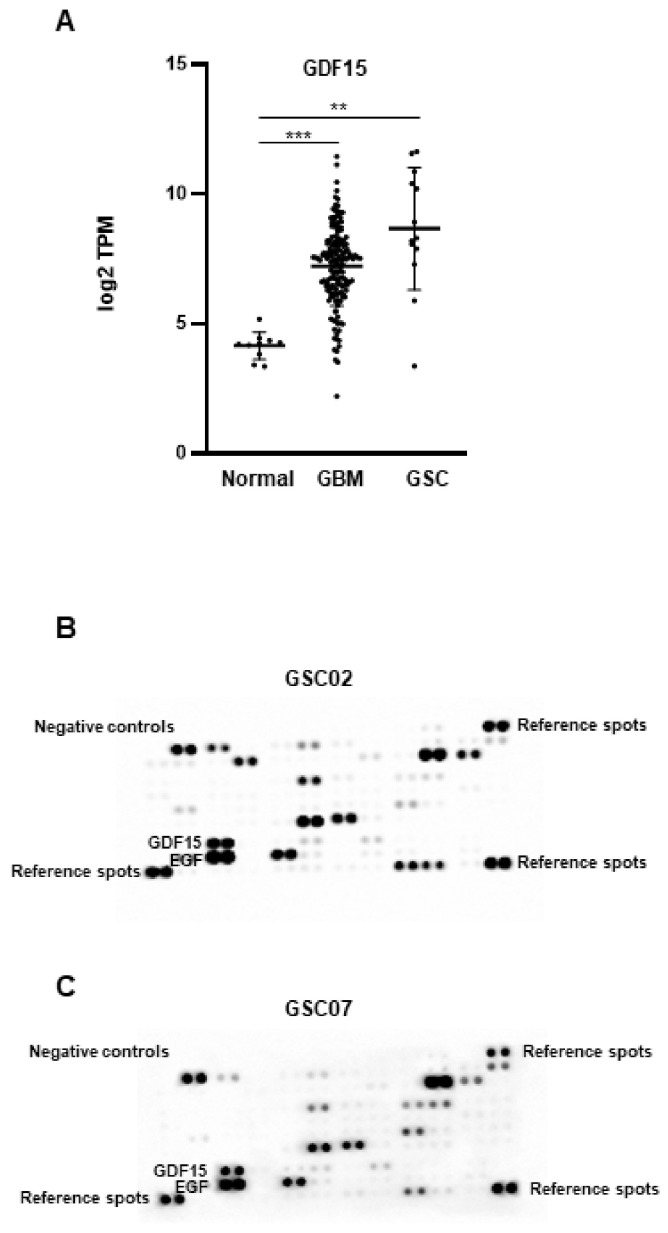
GDF15 expression in normal tissue, GBM, and GSCs. (**A**) The expression of GDF15 mRNA in normal tissue or GBM samples from the TCGA database was compared with the RNAseq data of GSCs derived from 13 human GBM biopsy specimens. The data were normalized with GAPDH expression. Pairwise t-tests were performed between the groups. Results are presented as means ± SD of log2 TPM. *** *p* < 0.001; ** 0.001 < *p* < 0.01. (**B**,**C**) GDF15 secretion by GSCs (GSC02, GSC07) was analyzed using the R&D Human XL Cytokine Array kit on 48 h conditioned media as described in “Methods”. EGF, which is added to the stem cell culture medium, can serve as a positive control. Images are representative of three independent experiments.

**Figure 2 cancers-16-00027-f002:**
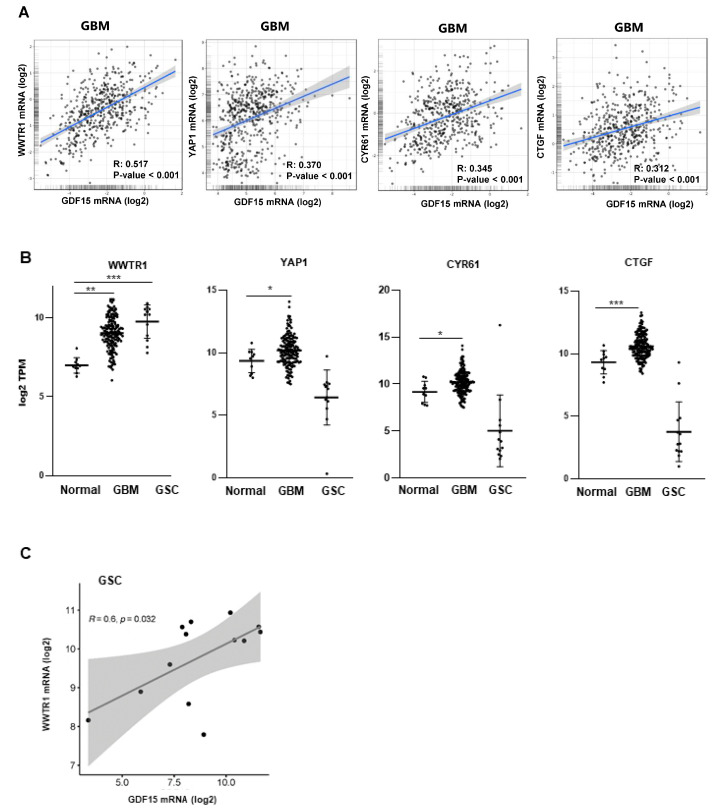
Correlations between the expression of GDF15 and the Hippo pathway genes. (**A**) The correlations between GDF15 expression and, respectively, WWTR1/TAZ, YAP1, CYR61, and CTGF were obtained by the co-expression analysis in Gliovis as described in “Methods” using the TGCA database. Values correspond to the Spearman correlation coefficient (R) and its associated *p*-value. (**B**) The expression of WWTR1/TAZ, YAP1, Cyr61, and CTGF mRNA in normal tissue or GBM samples from the TCGA database was compared with the RNAseq data of GSCs derived from 13 human GBM biopsy specimens. The data were normalized with GAPDH expression. Pairwise t-tests were performed between the groups. Results are presented as means ± SD of log2 TPM. *** *p* < 0.001; ** 0.001 < *p* < 0.01; * 0.01 < *p* < 0.05. (**C**) The correlation curve between GDF15 and WWTR1/TAZ in GSCs was performed with Srplot as described in “Methods” using the RNAseq data. The values correspond to the Spearman correlation coefficient (R) and its associated *p*-value.

**Figure 3 cancers-16-00027-f003:**
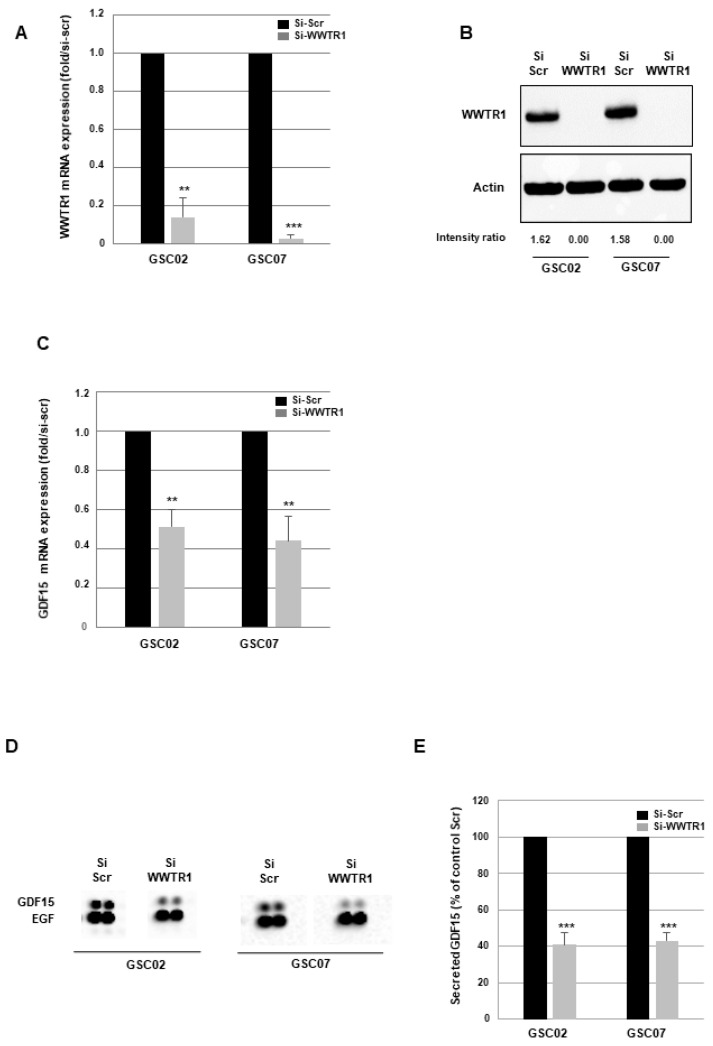
Blocking WWTR1/TAZ decreases GDF15 expression in GSCs. Primary GSC neurospheres from two different patients (GSC02, GSC07) expressing for 48 h a WWTR1/TAZ-specific siRNA (si-WWTR1/TAZ) or a scramble control (si-Scr) were used. WWTR1/TAZ (**A**) and GDF15 (**C**) mRNA expression was analyzed using real-time PCR using GAPDH expression for normalization. (**B**) WWTR1/TAZ protein expression was analyzed using western blot. (**D**,**E**) GDF15 secretion by the GSCs was analyzed using the R&D Human XL Cytokine Array kit on 48 h conditioned media as described in “methods”. (**E**) For the quantification, the average signal of the reference spots and the number of cells were used for normalization as described in “methods”. Images (**B**,**D**) are representative of three independent experiments. Quantifications of three experiments are presented as mean ± SD. *** *p* < 0.001; ** 0.001 < *p* < 0.01.

**Figure 4 cancers-16-00027-f004:**
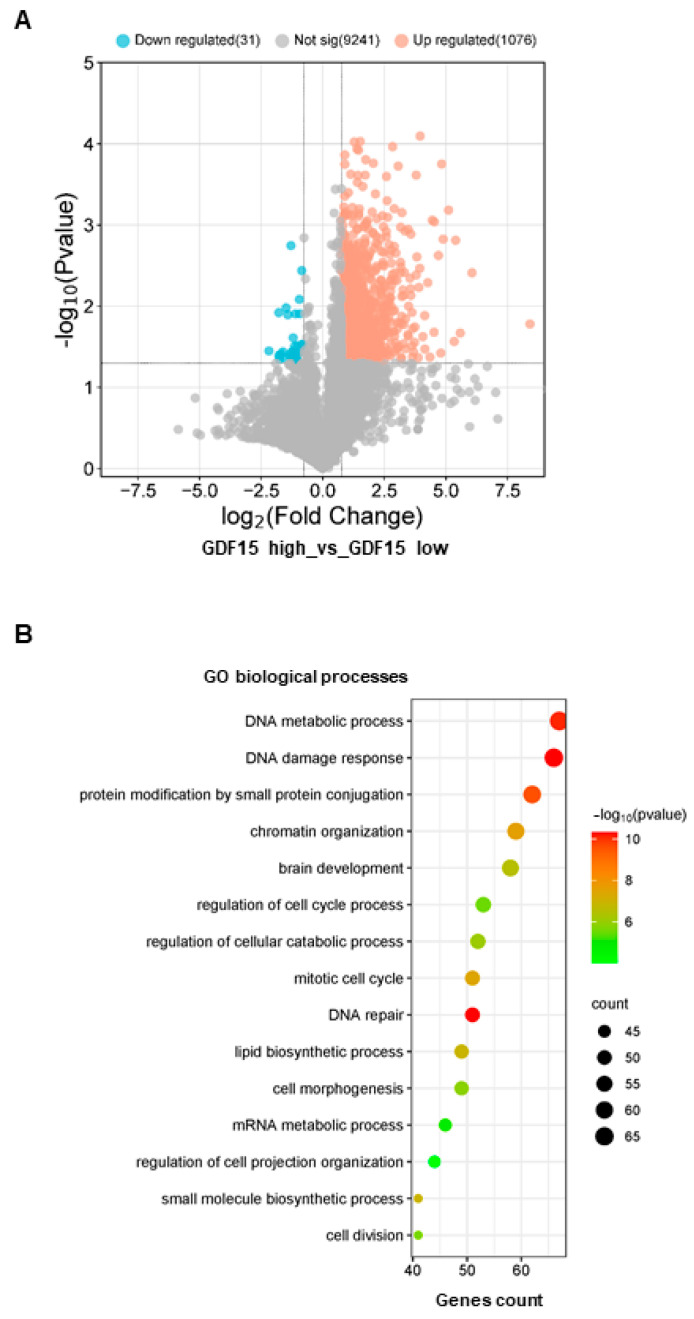
Genes and pathways associated with GDF15 high expression in GSCs. (**A**) Volcano plot showing the genes that are differentially expressed between GSCs with a strong expression of GDF15 (GDF15 high) compared with GSCs expressing GDF15 weakly (GDF15 low), performed as described in “Methods”. (**B**) Gene ontology analysis (biological processes) was performed on the genes significantly up-regulated in the high expressing GDF15 GSCs as described in “Methods”.

**Figure 5 cancers-16-00027-f005:**
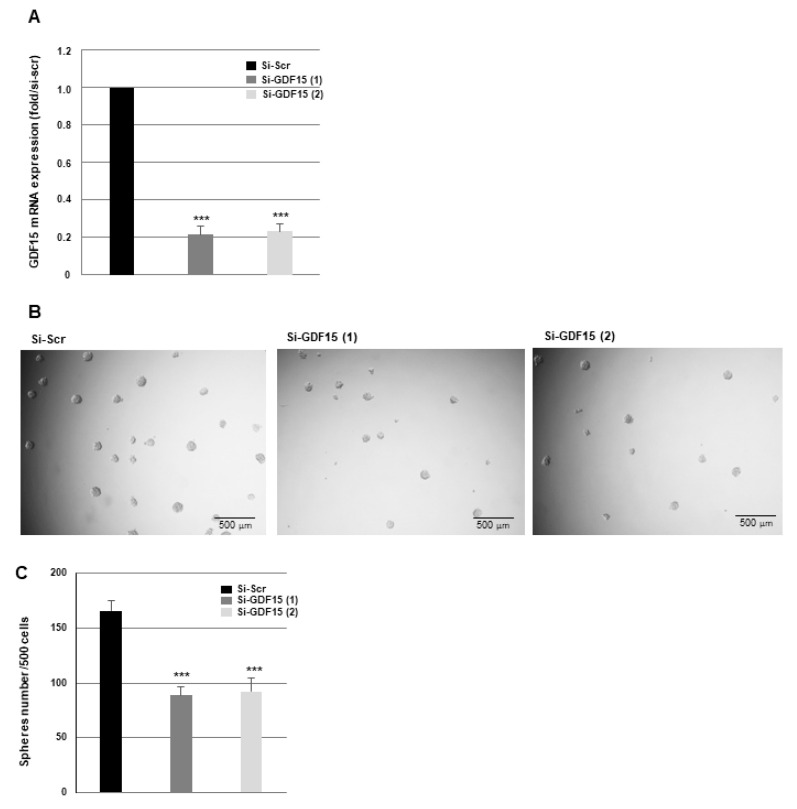
Down-regulation of GDF15 decreases the sphere-forming ability of GSCs. GSC02 expressing for 48 h specific siRNAs against GDF15 (si-GDF15(1), si-GDF15(2)) or a scramble control (si-Scr) were used for (**A**) GDF15 mRNA expression analyzed by real-time PCR, using GAPDH for normalization or (**B**,**C**) for neurosphere-forming analysis as described in “Methods”. Quantifications of three experiments are presented as mean ± SD. *** *p* < 0.001.

**Figure 6 cancers-16-00027-f006:**
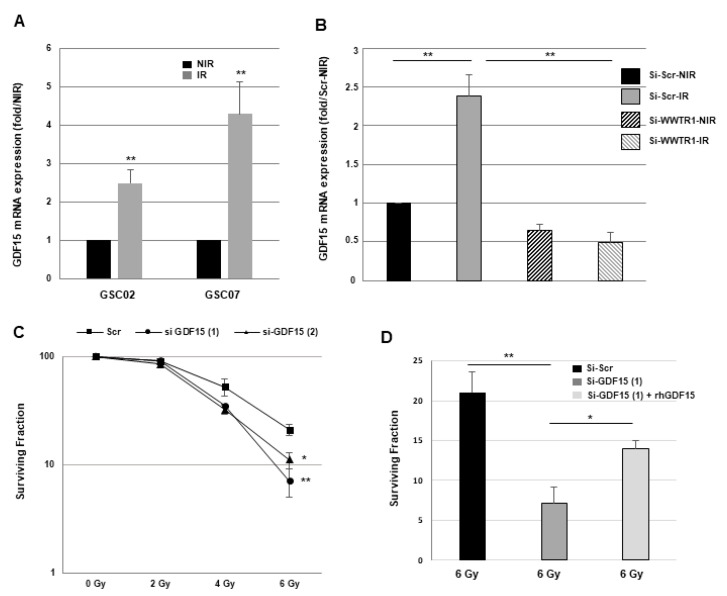
GDF15 mRNA expression is increased following IR in GSCs and down-regulation of GDF15 decreases radioresistance of GSCs. (**A**) Non-transfected GSCs or (**B**–**D**) GSC02 expressing for 48 h specific siRNAs against WWTR1/TAZ (si-WWTR1/TAZ), GDF15 (si-GDF15(1) or si-GDF15(2)), or a scramble control (si-Scr) were used. (**A**,**B**) GDF15 mRNA expression was analyzed under basal conditions (NIR) or 24 h after a single dose of IR (8 Gy) by real-time PCR, using GAPDH for normalization. (**C**) Neurosphere formation following increasing doses of x rays (0 to 6 Gy) was analyzed as described in “Methods”. (**D**) GSC02 expressing for 48 h a specific siRNA against GDF15 (si-GDF15(1)) or a scramble control (si-Scr) was used for the neurosphere-forming analysis following 6 Gy of IR as described in “Methods”. When indicated, rhGD15 (20 ng/mL) was added before IR. Quantifications of three experiments are presented as means ± SD. ** 0.001 < *p* < 0.01; * 0.01 < *p* < 0.05.

## Data Availability

The RNAseq data presented in this study are openly available in the SRA database under the reference PRJNA1020743.

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
