# Peer review of "Overexpression of Growth Differentiation Factor 15 in Glioblastoma Stem Cells Promotes Their Radioresistance"

_cancers, 2023, doi:10.3390/cancers16010027_

Round 1
Reviewer 1 Report
Comments and Suggestions for Authors
The authors present a study on "patient-derived GSC to demonstrate the overexpression of GDF15". However, they fail to substantiate such premise of their study. Specifically, there is no control sample for the expression level of GDF15 in Figure 1B, therefore it is impossible to conclude that the expression is higher compared to other samples (i.e., healthy). Similarly, Figure 2C also lacks controls. For this reason, the study is not adequate for publication.
Comments on the Quality of English Language
The quality of the English language is too low, so that it hinders the understanding of the study.
Author Response
"Please see the attachment."

Reviewer 2 Report
Comments and Suggestions for Authors
BENTABERRY-ROSA et al. demonstrated that overexpression of GDF15 is linked to radioresistance in glioblastoma stem cells. The authors explored the connection between GDF15 and TAZ expression in patient-derived GSCs through both computational analyses and functional assays. However, the overall research approach appears somewhat unclear and biased. I believe that the manuscript could benefit from addressing the following issues.
1. Figure 1C requires a control sample. It is challenging to determine strong signals or upregulation without control samples. The Proteome Profiler Array relies on relative signals generated by the ECL reagent and development time. The authors may consider adding control samples, such as differentiated glioma cells from GSC02 and GSC07.
2. In Figure 4D, does the blocking of WWTR1/TAZ result in a decrease in GDF15 alone within the Proteome Profiler Array panel? The Profiler Array encompasses multiple cytokines/proteins. It is crucial to demonstrate whether the silencing of WWTR1/TAZ specifically affects GDF15 to suggest the dependency of TAZ on GDF15 signaling. Researchers typically display whole membrane blots in this type of proteome array.
3. The authors propose that the WWTR1-GDF15 signaling pathway in GSCs mediates radioresistance. To directly prove this molecular axis, can the authors conduct a rescue experiment regarding the response to ionizing radiation (IR) using an ectopic GDF15 plasmid with and without si-WWTR1 (Figure 6)? There is no direct evidence supporting the correlation between radioresistance and WWTR1/GDF15.
4. According to the method section, Figure 6E appears to be a neurosphere-forming analysis, not a 3D clonogenic assay.
5. According to the legend for Figure 1C, the authors used media to assess GDF15 levels secreted from the tumor cells. A detailed method is needed to calculate the relative signal of GDF15 in the media relative to the cell number.
Comments on the Quality of English LanguageI see many typos and grammatical errors. The authors need to correct some errors.
Author Response
"Please see the attachment."

Reviewer 3 Report
Comments and Suggestions for Authors
The manuscript by Bentaberry-Rosa et al. reports that the cytokine GDF15 is overexpressed in glioblastoma stem cells (GSCs) and promotes radioresistance. In one patient-derived GSC culture it is shown that siRNA-mediated downregulation of GDF15 gives increased radiosensitivity. Furthermore, GDF15 is upregulated after irradiation in a manner dependent on the transcription factor WWTR1/TAZ. Although previous studies have shown that siRNA-mediated downregulation of GDF15 causes radiosensitization in cells from other cancer types, glioblastoma stem cells were not included in those studies. The results of this manuscript are thus interesting and might suggest GDF15 as a target for radiosensitization in glioblastoma. However, the manuscript appears too preliminary in its present form. Additional experiments are needed.
Major issues:
1. In most of the figures essential information is lacking about the experimental procedures. (E.g.: At what time after treatment were the samples harvested? When was the treatment done relative to the siRNA transfection? What was the siRNA concentration used for transfection? How were the gene expression data normalized in order to compare GSC & patient data? ......?)
2. The main conclusion is that GDF15 promotes radioresistance in glioblastoma stem cells. However, results supporting this claim are shown only for one GSC culture (GSC02; Fig 6D). Are similar radiosensitizing effects obtained after siRNA-mediated depletion of GDF15 in the other GSC cultures? (E.g. siGDF15 in GSC07?)
3. Another major conclusion is that GDF15 is overexpressed in GSCs. However, no comparison is made to other cultured cells (e.g. glioblastoma or normal cell lines).
4. No results (or discussion) are included to address potential mechanisms explaining the increased radiosensitivity upon GDF15 depletion. Why does depletion of GDF15 cause increased radiosensitivity in GSCs? Did any of the previous studies in other cancer types find anything about the mechanisms? Could the authors provide some data regarding this issue in the GSCs? (E.g. Fig 5 suggests that GDF15 level correlates with expression of DNA repair proteins, but it has not been tested whether GDF15 regulates DNA repair in the GSCs)
5. In fig 6B it is shown that siRNA-mediated depletion of WWTR1 completely suppresses the upregulation of GDF15 after irradiation. It would be interesting to know whether depletion of WWTR1 also causes radiosensitization, similarly as the depletion of GDF15 (siGDF15 in Fig 6D).
Minor issues:
6. There are many grammar errors throughout the manuscript (was/were, has/have, plural/ singular).
7. Figure 3: The two groups of GSCs are not defined. Are they present in Fig 1B? Could the correlation rather be shown by plotting the GDF15 levels versus the WWTR1/TZ1 levels , such as done in Fig 2A ?
8. It is not clearly described whether the results in Fig 6EF and Fig 6D are from the same experiments or not. -I guess that Fig 6EF shows the number of spheres for the non-irradiated samples (0 Gy) in Fig 6D?
9. Methods, line 140. How was the number of neurospheres/well measured?
( I could not open the supplement files and have therefore not reviewed them.)
Comments on the Quality of English LanguageThere are many grammar errors throughout the manuscript (was/were, has/have, plural/ singular).
Author Response
"Please see the attachment."

Round 2
Reviewer 1 Report
Comments and Suggestions for Authors
All the concerns were adequately addressed.
Author Response
We thank the reviewer 1 who found that all the points were adequately addressed in the first round
Reviewer 2 Report
Comments and Suggestions for Authors
The authors have addressed all the questions in this revision. I have no further concerns or comments.
Author Response
We thank the reviewer 2 who found that all the points were adequately addressed in the first round
Reviewer 3 Report
Comments and Suggestions for Authors
One of the main messages of the original manuscript, that GDF15 is overexpressed in glioblastoma stem cells, has been changed in the revised version. Proper controls had not been included in the original analysis. This shows that the original manuscript was submitted prematurely. The manuscript title therefore had to be changed in the revised version. Unfortunately the new manuscript title is "weaker" and less interesting than the original title.
Otherwise, the revised manuscript has been improved by inclusion of some more data and better explanation of the methods. However, there are still several unclear issues.
Specific comments:
- The normal tissue in fig 1 and fig 2 is not sufficiently described. It is just termed normal tissue. What kind of normal tissue is this? How many normal tissue samples are included in the analysis in the database?
-Fig 2c shows a range of GDF15 expression levels in GSCs. Where are GSC02 and GSC07 in this plot, do these GSCs express particularly high levels of GDF15?
- To what extent do the two selected GSCs, GSC02 and GSC07, express stem cell markers?
-A sentence is included on lines 70-73 saying that the13 GSCs have been previously characterized, but no reference is included to the previous study along with this statement.
-The two sentences next to each other in lines 258 -261 describe opposite conclusions. This is confusing. "Fig5B shows that blocking WWTR1.. reverse IR-induced GDF15 expression" ". .... indicating that GDF15 expression induced by IR is not dependent on WWTR1... "
-Line 270 "Finally, we examine the role of GD15 on the ability to form neurospheres (Fig 6A & 6B)". This statement is very confusing if the role of GDF15 on the ability to form neurospheres has already been described in the previous figure and sentences . (As I understand Fig 6A & 6B show neurosphere formation without IR & Fig 5DE show exactly the same assay with IR?)
-Fig S3: Information is lacking about the effect of siGDF15 at 0 Gy in GSC07. Furthermore, the extent of downregulation of GDF15 after siRNA transfection is not shown for GSC07.
-Fig S4: Information is lacking about the effects of si-WWTR1 at 0 Gy.
Round 3
Reviewer 3 Report
Comments and Suggestions for Authors
All my concerns regarding the results have been addressed satisfactorily.
However, I did not want the authors to change the title back to the original title. This was a misunderstanding. The title "GDF15 expression in glioblastoma stem cells contributes to radioresistance" is more correct considering the results & conclusions of the revised mansucript.
Author Response
According to the reviewer 3 we changed the title of the manuscript which is now
"GDF15 expression in glioblastoma stem cells contributes to radioresistance"